# A Symbolic Rule Integration Framework with Logic Transformer for Inductive Relation Prediction

## ABSTRACT

Relation prediction in knowledge graphs (KGs) aims at predicting missing relations in incomplete triples, whereas the dominant paradigm by KG embeddings has a limitation to predict the relation between unseen entities. This situation is called an inductive setting, which is more common in the real-world scenario. To handle this issue, implicit symbolic rules have shown great potential in capturing the inductive capability. However, it is still challenging to obtain precise representations of logic rules from KGs. The argument variability and predicate non-commutativity in symbolic rule integration make the modeling of component symbols difficult. To this end, we propose a novel inductive relation prediction model named SymRITa with a logic transformer integrating rules. SymRITa firstly extracts the subgraph, whose embeddings are captured by a graph network. Meanwhile, symbolic rule graphs in the subgraph can be generated. Then, the symbolic rules are modeled by a proposed logic transformer. Specifically, the input format based on the subgraph-based embeddings is to focus on the argument variability in symbolic rules. In addition, a conjunction attention mechanism in the logic transformer can resolve predicate non-commutativity in the symbolic rule integration process. Finally, the subgraph-based and symbol-based embeddings obtained from the previous steps are combined for the training regime, and prediction results as well as rules explaining the reasoning process are explicitly output. Extensive experiments on twelve inductive datasets show that SymRITa achieves outstanding effectiveness compared to state-of-the-art inductive baselines. Moreover, the logic rules with corresponding confidences provide an interpretable paradigm.

## CCS CONCEPTS

• **Computing methodologies → Knowledge representation and reasoning**.

## KEYWORDS

Inductive relation prediction, knowledge graph, first-order logic, logic transformer

**ACM Reference Format:**
Anonymous Author(s). 2018. A Symbolic Rule Integration Framework with Logic Transformer for Inductive Relation Prediction. In *Proceedings of Make sure to enter the correct conference title from your rights confirmation emai (Conference acronym 'XX)*. ACM, New York, NY, USA, 11 pages. https://doi.org/XXXXXXX.XXXXXXX

## 1 INTRODUCTION

In a structured storage scenario, knowledge graphs (KGs) possess numerous triples consisting of entities and relations, representing factual knowledge about the real world. They have been widely used in various downstream tasks, such as question answering [1, 10], information retrieval [6, 34], and text generation [15], etc. Existing KGs suffer from incompleteness according to the open-world assumption, which weakens the performance of downstream tasks. Therefore, some dominant methods aim at predicting missing entities and relations using the rich implicit structural information. There are many conventional methods for obtaining the relation and entity embeddings, such as TransE [3], R-GCN [28], CompGCN [33] and StAR [35].

Previous embedding-based methods are merely designed for a transductive setting. However, in the real-world scenario, the relation prediction is always implemented in an inductive setting, which predicts the relation between two unseen entities in the test set. The above mentioned methods are not suitable for this situation, because they require to retrain the whole model for unseen entities and lack an inductive ability in applications. With the development of large language models (LLMs), some recent methods try to implement reasoning by LLMs [44, 48], while they require massive manual prompts and the performance does not meet the desired outcome, especially on unseen out-of-distribution samples [2]. Based on the circumstance, some models [19, 31] provide a thought of capturing inductive ability by mainly taking account of the topological representations in KGs. Despite these methods, they still neglect the entity independent information that is critical to provide inductive capability for the model.

Some methods have illustrated that the symbolic rule, specifically the first-order logic rule [4, 16], can provide entity independent information by its format and capture the inductive ability [43] during reasoning in KGs. For example, in Figure 1, with the information of the symbolic rule:

$$writtenBy(X, Y) \leftarrow prequelOf(X, Z) \wedge writtenBy(Z, Y),$$

the relation *writtenBy* between the entities *George R. R. Martin* and *A Game of Thrones* in the test set can be predicted without retraining the whole model. However, there are still some important issues in integrating the symbolic rules into the inductive model:

**(1) Argument variability.** Integrated rules in KGs follow an argument variability of the logic [11], which is essential for obtaining entity independence for the model. For instance, in Figure 1, the entities *Harry Potter and the Philosopher's Stone* and *J. K. Rowling* are generalized to arguments $X$ and $Y$ respectively. The arguments are variable in the symbolic rules, which means that if the arguments in $writtenBy(X, Y) \leftarrow prequelOf(X, Z) \wedge writtenBy(Z, Y)$ are changed from $X, Y, Z$ to $A, B, C$, the semantics of the rule does not change. Although existing methods introduce implicit rules, they still have difficulties in modeling the information of arguments. **(2) Predicate non-commutativity.** The symbolic rules

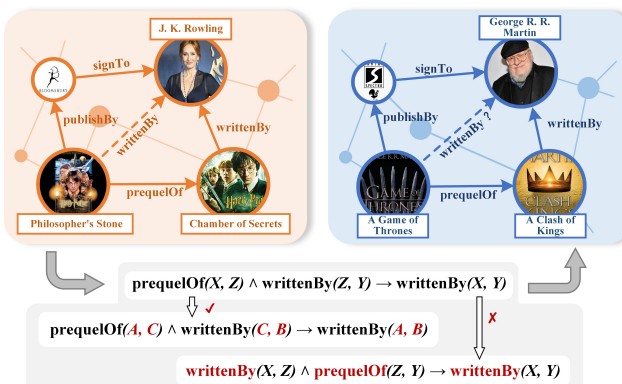

**Figure 1: The issues of integrating the symbolic rule into inductive relation prediction.**

have a predicate non-commutativity, which means that the semantics of logic rules depends on the order of predicates [23]. As shown in Figure 1, the atoms $prequelOf(X, Z)$ and $writtenBy(Z, Y)$ are connected by the conjunction symbol ∧ in the rule. The predicates $prequelOf$ and $writtenBy$ in the two atoms can not be switched without the arguments, for it will transfer to different semantics, even wrong semantics. For example, if we exchange predicates $prequelOf$ and $writtenBy$, the argument $Z$ will be instantiated to an exact person, but a person will not be a prequel of another thing. The rule after exchanging is obviously unreasonable during reasoning, which means the order of predicates is critical for representing the semantics of the logic.

To address the above issues, we propose a **Sym**bolic **R**ule **I**ntegration model with logic **Tra**nsformer for inductive relation prediction named SymRITa. In SymRITa, the logic transformer module is proposed to obtain embeddings of symbolic rules during reasoning. In detail, we firstly extract the enclosing subgraph among the target triple, which performs as the foundation to extract implicit rules. Following the format of the first-order logic rule, we extract them within a maximum preset length as the skeletons. Secondly, we implement the subgraph modeling and obtain the subgraph-based embeddings of entities and relations through a graph network based on the graph topological information. Then, we propose a rule integration strategy to integrate the implicit logic and obtain symbol-based embeddings for inductive ability. The inputs of logic transformer are from the subgraph-based relation embeddings, which aims at the argument variability in the integrating. As for the predicate non-commutativity, an unscathed conjunction attention mechanism is designed for the logic transformer, providing comprehensive information for symbolic rule integration. Finally, SymRITa implements the inductive relation prediction by combining subgraph-based and symbol-based embeddings and jointly trains the model. The symbol-based embedding is obtained by each symbolic rule and its corresponding confidence, which illustrates the explanation of the reasoning process.

Our main contribution are in the following three folds:

- A novel framework named SymRITa integrates the symbolic rules into the continuous neural network model to solve the inductive relation prediction. To the best of our knowledge, it is the first method to model symbol-based embeddings

and combine them with subgraph-based embeddings for obtaining the inductive ability.
- In order to model the discrete symbolic rules for resolving entity independence in the inductive setting, we propose a logic transformer and design a conjunction attention mechanism to satisfy the argument variability and strengthen the predicate non-commutativity of first-order logic rules extracted from KGs.
- Experiments of the relation prediction on twelve inductive datasets verify the effectiveness of SymRITa compared to the latest inductive methods. Meanwhile, SymRITa obtains first-order logic rules for the explanation of reasoning process.

The rest of this paper is organized as follows. Section 2 introduces the related work of inductive relation prediction and symbolic reasoning. The preliminary given in Section 3 illustrates the first-order logic rule from KGs and the task definition. The demonstration of our method SymRITa is in Section 4. Section 5 carries out extensive experiments and analyzes the effectiveness of SymRITa. Finally, we conclude this paper and discuss the future work in Section 6.

## 2 RELATED WORK

In this section, we introduce some recent works about inductive relation prediction and symbolic reasoning related to SymRITa.

### 2.1 Inductive Relation Prediction

For the inductive relation prediction task on KGs, we divide them into two main categories: rule-based and graph-based.

**Rule-based** methods extract logic rules from KGs enhancing the inductive ability instead of fusing with external information [38, 39]. Associated methods mine closed-paths rules from enumerated candidates in KGs by statistical indicators, such as *head coverage* in AMIE [12, 13], *pointwise mutual information* in SHER-LOCK [29], and other indicators in early works [20, 24]. RuleN [21] extends AMIE with a fine-grained evaluation, which extracts more rules with high qualities. Considering the lacking scalability of these associated methods, some other differential methods solve the inductive problem by learnable weights for rules during the reasoning process in KGs. NeuralLP [43] learns the confidences and the structures of logic rules through an end-to-end model, which consists of a controller system implemented by a recurrent neural network (RNN). DRUM [27] extends NeuralLP with the Bi-RNN to extract logic rules with flexible length. Both of the methods are based on the framework named TensorLog [7], which transfers the discrete rule extracting to continuous learnable parameters. Nevertheless, these methods are still struggling with inadequate inductive reasoning performance based on the framework with limited inductive ability and high resource consumption.

**Graph-based** methods try to take account of the topological information of KGs to get more sufficient inductive ability. GraIL [31] mines the inductive ability of KGs from extracted subgraphs and models them by a graph neural network (GNN). CoMPILE [19] is proposed based on GraIL, considering the implicit inductive ability by interacting messages between edges and entities. TACT [5] extends GraIL by a relational correlation network to supply different patterns of topological information between relations.

RED-GNN [49] models the KG by a relational directed graph, in order to capture the local evidence of the KG for inductive reasoning. However, these methods merely utilize the topological information without taking advantage of the implicit semantics.

Distinguished from the previous methods, SymRITa combines the topological structure with symbolic rules, which is significant for improving the inductive ability of the model.

## 2.2 Symbolic Reasoning

Symbolic reasoning aims to integrate symbolic formulas into the reasoning tasks, which would not only develop the performance but also the interpretability of the reasoning process. For the visual problems, NS-VQA [45] incorporates symbolic structure as prior knowledge to a visual question answering task. It recovers a structural scene representation from the image and a symbolic program trace from the question, and then executes the program to obtain an answer. For the logical question answering task, LReasoner [36] proposes a context extension framework based on logical equivalence laws to capture symbolic logic from the text. As for the graph networks, SGR [18] performs reasoning over a group of symbolic nodes whose outputs explicitly represent different properties of semantics in a prior graph. It learns shared symbolic representations for domains or datasets with the different label set.

Symbolic reasoning is applied to text, images, graphs and multi-modal scenarios. For solving the inductive setting and getting the inductive capability, we innovatively introduce the symbolic reasoning into the inductive relation prediction.

## 3 PRELIMINARY

### 3.1 Inductive Relation Prediction

Inductive relation prediction in KGs aims to predict the relation between two unseen entities. A target triple is denoted as $(h, r_T, t)$ in the train KG $G = \{R, E, T\}$, in which $r_T \in R$ is the target relation. $h, t \in E$ are head and tail entities, respectively. $R$ and $E$ are sets of relations and entities in $G$, and $T \subseteq E \times R \times E$ is the set of triples. Inductive relation prediction intends to predict if the triple $(h', r_T, t')$ is valid with two unseen entities $h'$ and $t'$ in a testing KG $G' = \{R, E', T'\}$. $G$ and $G'$ share the same relation set. However, entities in $G$ and $G'$ are disjoint, i.e. $E' \cap E = \varnothing$. For clarity, we summarize important symbols in Table 1.

### 3.2 Symbolic Rule in KGs

The symbolic rule [22] learned from KGs is a first-order logic Horn rule [26], which consists of a head atom and a series of atoms as the body. Here is a first-order logic rule with length $N$:

$$\beta \overbrace{r_T(X, Y)}^{head} \leftarrow \overbrace{r_1(X, Z_1) \wedge r_2(Z_1, Z_2) \wedge ...r_N(Z_{N-1}, Y)}^{body}. \quad (1)$$

in which $r_1, r_2, ..., r_N, r_T$ are predicates, represented as relations in KGs. $X, Z_1, Z_2, ..., Z_{N-1}, Y$ are arguments generalized from entities in KGs. The body atoms $r_1(X, Z_1), \cdots, r_N(Z_{N-1}, Y)$ are connected by a conjunction symbol $\wedge$ and point to the head by an implication symbol $\leftarrow$. In the rule (1), an atom contains two arguments and adjacent atoms share the same argument. During inference, arguments are instantiated to entities $x, z_1, z_2, \ldots, z_{N-1}, y$ and form a

**Table 1: Important symbols and their descriptions.**

| Symbol | Description |
|---|---|
| $a_T = (h, r_T, t)$ | Target triple to be predicted |
| $G, G'$ | Train and test KGs for inductive prediction |
| $\mathcal{S}_T$ | The subgraph of $a_T$ |
| $\boldsymbol{v}^{(L_1)}$ | Embedding vector of node $i$ at layer $L_1$ |
| $L_{max}$ | Max length of the relational paths |
| $\mathcal{F} = \{f'_1, f'_2, \cdots, f'_n\}$ | Symbolic rules extracted from $\mathcal{S}_T$ |
| $\boldsymbol{F}$ | The embedding vector of rules in $\mathcal{F}$ |
| $\boldsymbol{F_o}$ | The embedding vector of rules after adding positional information |
| $\boldsymbol{S}$ | Subgraph-based embedding |
| $\boldsymbol{H}^{(L_2)}$ | Output of logic transformer after $L_2$ layers |
| $\boldsymbol{P}$ | Symbolic-based embedding |

closed path. In the KG, we can use the target relation to simulate the head predicate of the rule. Also, $\beta \in [0, 1]$ can be used to represent the *confidence* of the rule. According to [43], these are important components in the first-order logic rules extracted from KGs.

## 4 METHODOLOGY

In this section, we demonstrate SymRITa with the help of the framework in Figure 2. SymRITa is in three parts.

### 4.1 Subgraph Modeling

SymRITa extracts the enclosing subgraph $\mathcal{S}_T$ based on the target triple $a_T = (h, r_T, t)$ from $G$. The entities in $\mathcal{S}_T$ are selected by the intersection of $q$-hop undirected neighborhoods of the target triple. The triples among the selected entities conduct the subgraph $\mathcal{S}_T$. Furthermore, $\mathcal{S}_T$ is used for labeling nodes by a double radius vertex labeling scheme [47]. The labeling process initializes the entities in the subgraph, and provides the initialized vector with the inductive ability by the topological structure [31]. Then, we initialize the node $i$ with a vector [one-hot$(d(i, h))\oplus$one-hot$(d(i, t))] \in \mathbb{R}^{(2q+2)}$ as the node feature, where $\oplus$ is the concatenation operation of vectors. $d(\cdot)$ generates the shortest topological distance between two entities.

For subgraph modeling, we employ a graph convolutional network (GCN) [28] based approach with relation-aware and triple-aware attention to obtain entity and relation embeddings in a $\mathcal{S}_T$. In detail, the update process of the entity embeddings is:

$$\boldsymbol{v}_i^{(l+1)} = \phi\left(\sum_{r \in R, j \in \mathcal{N}_i^r} \alpha_{i,r}^{(l)} \mathbf{W}_r^{(l)} \boldsymbol{v}_j^{(l)} + \mathbf{W}_0^{(l)} \boldsymbol{v}_i^{(l)}\right), \quad (2)$$

$$\boldsymbol{\omega}_{i,r}^{(l)} = \sigma_1(\mathbf{W}_1^{(l)} [\boldsymbol{v}_i^{(l)} \oplus \boldsymbol{v}_j^{(l)} \oplus \boldsymbol{r}]), \quad (3)$$

$$\alpha_{i,r}^{(l)} = \sigma_2(\mathbf{W}_2^{(l)} [\boldsymbol{\omega}_{i,r} \oplus \boldsymbol{r}_T]), \quad (4)$$

where we use $\boldsymbol{v}_i^{(l+1)}$ to be the embedding of node $i$ in $(l+1)-th$ layer from $L_1$ layers during message passing. $\phi$ denotes the multilayer perceptron (MLP) operation to aggregate the neighbor information. $\mathcal{N}_i^r$ collects the neighbors of node $i$ by relation $r$. $\mathbf{W}_r^{(l)}$ and $\mathbf{W}_0^{(l)}$ are transformation matrices to update the message from layer $l$ to $l + 1$. $\alpha_{i,r}^{(l)}$ is the triple-aware attention weight of neighbor triple connected by relation $r$. $\boldsymbol{\omega}_{i,r}$ denotes the triple-aware embedding and is further used for calculating the triple-aware attention weight

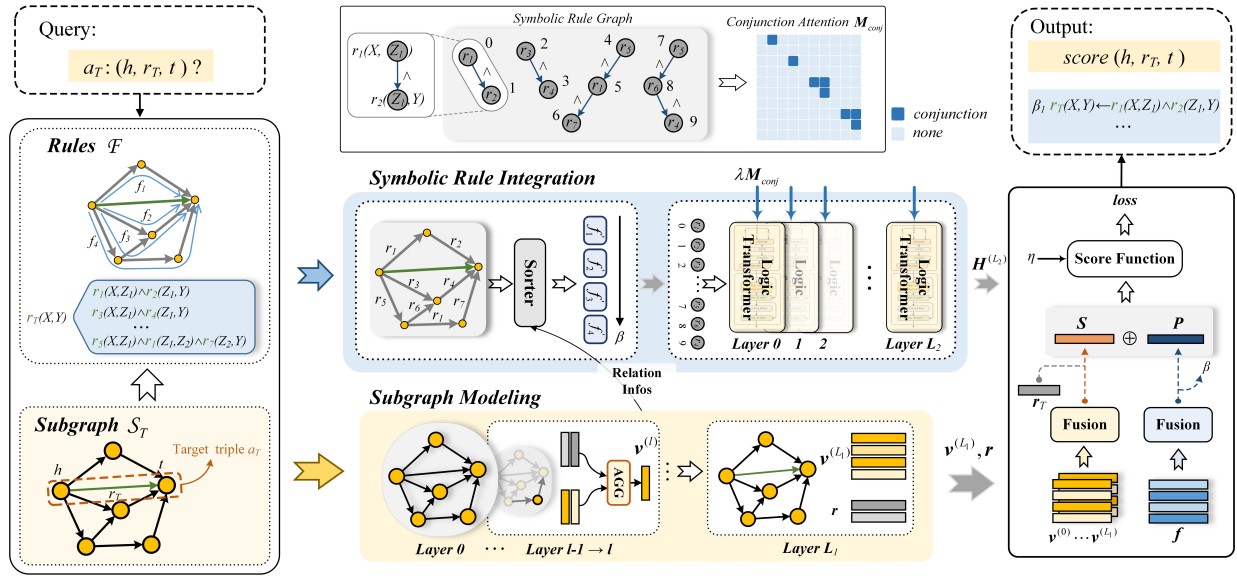

**Figure 2: The overall framework of SymRITa. It firstly obtains subgraphs given a target triple, and the embeddings of relations and entiites in the subgraph. Then, the relation embeddings are the input of logic transformer to integrate symbolic rules and obtain rule-based embeddings. Finally, the subgraph-based embeddings and rule-based embeddings are used for training and prediction process.**

$\alpha_{i,r}^{(l)} \cdot r_T$ is the embedding of the target relation $r_T$. $\sigma_1$ and $\sigma_2$ are activation functions, and $\mathbf{W}_1^{(l)}, \mathbf{W}_2^{(l)}$ are transformation matrices.

## 4.2 Symbolic Rule Integration

As for the symbol components in rules, we design a strategy to embed them for enhancing the inductive ability during reasoning. This is the main part of our SymRITa, and we propose a graph transformer-based method to model the rules in subgraphs.

*4.2.1 Logic Extraction and Sorter.* We construct the implicit logic rules from the extracted subgraphs. From the format of the first-order logic rule shown as Eq. (1), the relational paths in a subgraph are significant parts of the implicit logics in a KG. If we get the relational paths from head to tail of the target triple $a_T$, we could obtain the skeletons of logic rules in $\mathcal{S}_T$. Specifically, we use the breadth first search (BFS) algorithm [8] for extracting every topological relational path whose length is not longer than $L_{max}$ from $h$ to $t$ in $\mathcal{S}_T$. The set of $n$ logic rules among $a_T$ is denoted as $\{f_1, \cdots, f_n\}$. If $L_{max}$ is preset as 3, $\mathcal{S}_T$ owns 4 relational paths from Figure 2 to represent the implicit logic rules for reasoning the relation between $h$ to $t$. According to Eq. (1), the arguments (e.g. $X, Y, Z$) and conjunction symbols (i.e. $\wedge$) are significant in rules, so the predicates in a single extract relational path should be in order, for instance, the relation order of $f_1 = \{r_1, r_2\}$ in $\mathcal{S}_T$. The information of arguments and the semantics of conjunction symbol in the rules is fused in SymRITa, which will be demonstrated in the following sections.

Then, the symbolic rules are sorted by their semantics in order to match the significance of the corresponding rule during reasoning. Based on a rule learning work [42], we design a strategy to calculate the semantic similarity between $r_T$ and the rule body $f_k$, in which $r_T$ and $f_k = \{r_{k,1}, \cdots, r_{k,l_k}\}$ connect the same $h$ and $t$. $l_k$ is the

number of relations in the format of $f_k$. In SymRITa, we denote the similarity to be $\theta_k$:

$$\theta_k = \varphi(f_k) \cdot r_T = \big( \sum_{r_i \in f_k} r_i \big)^{\top} r_T \qquad (5)$$

where $\varphi$ is an aggregator for combining all the relations in $f_k$. The value of the semantic similarity $\theta_k$ indicates the rationality of a first-order logic rule body $f_k$. We sort them by value of $\theta$ in a descending order, and represent them as a sequence $\mathcal{F} = \{f'_1, f'_2, \cdots, f'_k, \cdots, f'_n\}$, where $n$ indicates the number of bodies in $\mathcal{F}$. This process can determine the relation sequence in symbolic rule integration, which is a reference for the positional information sent to the next part named logic transformer.

*4.2.2 Symbolic Rule Graph and Logic Transformer.* As a critical part of the rule integration, we propose a module to embed not only the semantics of relations in rules, but also the semantics of arguments and conjunction symbols. Inspired by the work of graph transformer [46], the architecture of logic transformer is devised. From the previous subsection, SymRITa treats the relations to be the predicates in the first-order rules. Because of the entity independence, we should not embed the exact entities in $\mathcal{S}_T$. Therefore, we design a module aiming at different arguments in the adjacent atoms connected by conjunction symbols.

As described in Section 3, when considering the arguments, they are always fused with the semantics of predicates to construct the body atoms in a first-order logic rule. For example, the head atom consists of the target relation $r_T$ and exact arguments $X$ and $Y$, while the body relations of the rule are connected by conjunctions $\wedge$ and arguments with variability. It can be structured to a directed graph, which is shown in the gray block in Figure 2 named symbolic rule graph. The nodes refer to atoms in the body (gray nodes), and the directed edges (blue arrows) are conjunction symbols connecting

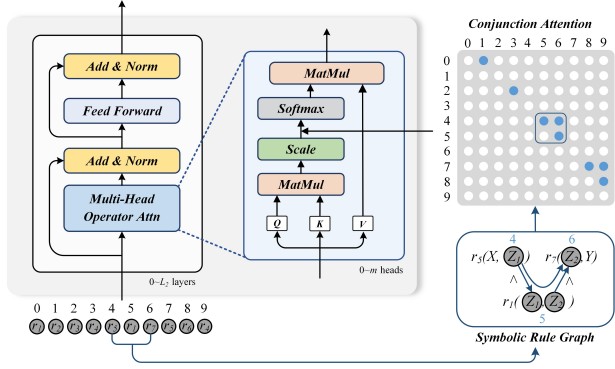

**Figure 3: The components of logic transformer.**

the same argument in adjacent atoms. Directed symbolic rule graphs follow the perspective of the reasoning process by the first-order logic rules as Eq. (1).

The input of the logic transformer for each $\mathcal{S}_T$ consists of the relation embeddings from elements in $\mathcal{F}$, which satisfies the requirement of the model for argument variability:

$$F = [\overbrace{r_{1,1}; r_{1,2}; \cdots ; r_{1,l_1}}^{f_1'}; \cdots ; \overbrace{r_{n,1}; r_{n,2}; \cdots ; r_{n,l_n}}^{f_n'}]. \quad (6)$$

According to the sorting result of the sorter, the position number of a node is determined by its position in the body $f_k'$ and the position of $f_k'$ in $\mathcal{F}$. Generally, we number the relations in $F$ and consider them to be the position information of the logic transformer. Therefore, the positional embeddings are added into the rule embeddings, which is denoted as:

$$F_o = F + \text{PosEmb}(F) \quad (7)$$

where $\text{PosEmb}(\cdot)$ provides a positional embedding to each node in the input from directed graphs.

Afterwards, we feed the overall node embedding $F_o$ into the logic transformer. As shown in Figure 3, the self-attention module is implemented by three matrices $Q, K$ and $V$, which indicate the query, key and value in the transformer:

$$Q^i, K^i, V^i = F_o W_q^i, F_o W_k^i, F_o W_v^i, \quad (8)$$

where $W_q^i, W_k^i$ and $W_v^i$ are trainable parameter matrices of the $i - th$ head in the logic transformer, which project the input $F_o$ to the corresponding representations $Q^i, K^i$ and $V^i$. Then, the attention is calculated by the matrices and the conjunction attention:

$$A^i = \frac{Q^i K^{i^\top} + \lambda M_{conj}}{\sqrt{d_K}}, \quad (9)$$

$$\text{Attn}^i(F_o) = \text{softmax}(A^i)V^i, \quad (10)$$

in which $M_{conj}$ is the conjunction attention we propose to introduce the operator information to the rule integration. $\lambda$ is a hyper-parameter adjusting the weight of the conjunction attention. $d_K$ is the scaling factor. We use the adjacent matrix $M_{conj}$ to represent the symbolic rule graph. The element in $M_{conj}$ is defined as $M_{conj}[i, j] = 1$ when the nodes $i$ and $j$ have a directed edge or a path. For example, in Figure 3, the position of the nodes are from the symbolic graphs corresponding to the input of the logic transformer, which is represented as Eq. (6). For the rule

body $r_5(X, Z_1) \wedge r_1(Z_1, Z_2) \wedge r_7(Z_2, Y)$, the elements of $M_{conj}[4, 5]$, $M_{conj}[4, 6]$ and $M_{conj}[5, 6]$ are set to 1. These three elements in $M_{conj}$ simultaneously represent the directed graph in Figure 3.

After $L_2$ layers, the representations of the hidden states $H^{(L_2)}$ are treated as the output of the logic transformer:

$$H^{(L_2)} = [\overbrace{h_{1,1}; h_{1,2}; \cdots ; h_{1,l_1}}^{f_1'}; \cdots ; \overbrace{h_{n,1}; h_{n,2}; \cdots ; h_{n,l_n}}^{f_n'}]. \quad (11)$$

## 4.3 Training and Prediction

After modeling the topological and logical information by subgraphs and symbolic rules, we fuse the embeddings and use the representations to train the model and implement the reasoning process. Specifically, the overall subgraph-based embedding $S$ is denoted as:

$$v_T^{(L_1)} = \frac{1}{|\mathcal{V}_T|} \sum_{i \in \mathcal{V}_T} v_i^{(L_1)}, \quad (12)$$

$$S = v_T^{(L_1)} \oplus v_{a_T}^{(L_1)} \oplus r_T, \quad (13)$$

in which $v_{a_T}^{(L_1)}$ indicates the concatenated embeddings of entities. $\mathcal{V}_T$ is the set of nodes in $\mathcal{S}_T$. In this process, we apply the JK-connection [31, 41] in getting the subgraph-based embedding $v_{a_T}^{(L_1)} = \left[ \bigoplus_{l=1}^{L_1} (v_h^{(l)} \oplus v_t^{(l)}) \right]$.

For the symbol-based embedding, we use the output $H^{(L_2)}$ of the logic transformer to obtain the overall symbolic rule representation $P$ in $\mathcal{S}_T$ and the confidence $\beta_k$ of symbolic rule $f_k'$:

$$P = \sum_{k=1}^n \beta_k f_k' = \sum_{k=1}^n \beta_k \left( \sum_{j=1}^{l_k} h_{k,j} \right), \quad (14)$$

$$\beta_k = \text{softmax}(f_k', r_T) = \frac{\exp(f_k'^\top r_T)}{\sum\limits_{f_p' \in \mathcal{F}} \exp(f_p'^\top r_T)}, \quad (15)$$

where $n$ is the number of rules within length $L_{max}$ in $\mathcal{S}_T$. Eventually, we evaluate the target triple by the link prediction, so the score of $a_T$ combining subgraph and symbolic information is denoted as:

$$score(a_T) = W_{score}[S \oplus P], \quad (16)$$

where $W_{score}$ refers to the weight matrix. Then the training process is implemented by a margin-based loss [37] to distance scores of positive and negative samples:

$$\mathcal{L} = \sum_{a_T \in \mathcal{E}} \max(0, \eta + score(a_T^-) - score(a_T^+)), \quad (17)$$

in which $\mathcal{E}$ refers to the set of triples in $G$, and $\eta$ is the margin representing the distance. $a_T^+$ indicates the positive sample and $a_T^-$ refers to the negative one. As for symbolic rules, we randomly replace predicates in $\mathcal{S}_T$ to construct negative samples.

## 5 EXPERIMENTAL RESULTS

In this section, we firstly introduce datasets, baselines, experiment settings, and details. Secondly, to verify the effectiveness of SymRITa, we implement comparison experiments on the inductive relation prediction task. Then, we use ablation studies, weight analysis and other studies to comprehensively demonstrate the performance.

**Table 2: Comparison of AUC-PR (%) and Hits@10 (%) results on inductive benchmarks from WN18RR, FB15K-237 and NELL-995. Results are from [25] and [14]. The optimal and suboptimal values are marked in bold and underline respectively.**

| Metric | Category | Method | WN18RR | | | | FB15K-237 | | | | NELL-995 | | | |
|---|---|---|---|---|---|---|---|---|---|---|---|---|---|---|
| | | | v1 | v2 | v3 | v4 | v1 | v2 | v3 | v4 | v1 | v2 | v3 | v4 |
| AUC-PR | Rule-based | RuleN | 90.26 | 89.01 | 76.46 | 85.75 | 75.24 | 88.70 | 91.24 | 91.79 | 84.99 | 88.40 | 87.20 | 80.52 |
| | | Neural-LP | 86.02 | 83.78 | 62.90 | 82.06 | 69.64 | 76.55 | 73.95 | 75.74 | 64.66 | 83.61 | 87.58 | 85.69 |
| | | DRUM | 86.02 | 84.05 | 63.20 | 82.06 | 69.71 | 76.44 | 74.03 | 76.20 | 59.86 | 83.99 | 89.71 | 85.94 |
| | Graph-based | GraIL | 94.32 | 94.18 | 85.80 | 92.72 | 84.69 | 90.57 | 91.68 | 94.46 | 86.05 | 92.62 | 93.34 | 87.50 |
| | | TACT | 95.43 | 97.54 | 87.65 | 96.04 | 83.15 | 93.01 | 92.10 | 94.25 | 81.06 | 93.12 | 96.07 | 85.75 |
| | | CoMPILE | 98.29 | 99.36 | 93.60 | **99.51** | 83.06 | 90.21 | 93.12 | 93.24 | 82.39 | 93.30 | 95.71 | 52.98 |
| | | LogCo | 99.43 | 99.45 | 93.99 | 98.75 | 89.74 | 93.65 | 94.91 | 95.26 | 91.24 | 95.96 | 96.28 | 87.81 |
| | | RMPI-NE | 95.09 | 95.43 | 88.58 | 94.82 | 85.22 | 92.08 | 91.77 | 92.27 | 81.07 | 93.64 | 94.99 | 88.82 |
| | | RMPI-NE-TA | 95.05 | 95.48 | 88.35 | 94.87 | 85.90 | 92.96 | 92.72 | 93.33 | 77.89 | 94.31 | 95.89 | 72.34 |
| | Ours | SymRITa | **99.58** | **99.46** | **94.02** | 98.77 | **89.77** | **93.74** | **95.19** | **95.28** | **92.59** | **95.99** | **96.94** | **91.91** |
| Hits@10 | Rule-based | RuleN | 80.85 | 78.23 | 53.39 | 71.59 | 49.76 | 77.82 | **87.69** | 85.60 | 53.50 | 81.75 | 77.26 | 61.35 |
| | | Neural-LP | 74.37 | 68.93 | 46.18 | 67.13 | 52.92 | 58.94 | 52.90 | 55.88 | 40.78 | 78.73 | 82.71 | 80.58 |
| | | DRUM | 74.37 | 68.93 | 46.18 | 67.13 | 52.92 | 58.73 | 52.90 | 55.88 | 19.42 | 78.55 | 82.71 | 80.58 |
| | Graph-based | GraIL | 82.45 | 78.68 | 58.43 | 73.41 | 64.15 | 81.80 | 82.83 | **89.29** | 59.50 | 93.25 | 91.41 | 73.19 |
| | | RED-GNN | 79.90 | 78.00 | 52.40 | 72.10 | 48.30 | 62.90 | 60.30 | 62.10 | **86.60** | 60.10 | 59.40 | 55.60 |
| | | TACT | 84.04 | 81.63 | 67.97 | 76.56 | 65.76 | 83.56 | 85.20 | 88.69 | 79.80 | 88.91 | 94.02 | 73.78 |
| | | CoMPILE | 81.91 | 76.64 | 57.35 | 71.80 | 62.20 | 82.01 | 84.67 | 87.44 | 58.33 | 88.86 | 93.63 | 60.81 |
| | | LogCo | 90.16 | 86.73 | 68.68 | 79.08 | 73.90 | 84.21 | 86.47 | 89.22 | 61.75 | 93.48 | 94.44 | 80.82 |
| | | RMPI-NE | 89.63 | 83.22 | 70.33 | 79.81 | 70.00 | 82.85 | 83.18 | 86.52 | 60.50 | 94.01 | 91.78 | 84.27 |
| | | RMPI-NE-TA | 87.77 | 82.43 | 73.14 | 81.42 | 71.71 | 83.37 | 86.01 | 88.69 | 60.50 | 93.49 | 95.30 | 66.42 |
| | Ours | SymRITa | **91.22** | **88.32** | 73.22 | **81.67** | 74.87 | **84.41** | 87.11 | 88.97 | 64.50 | **94.22** | **95.43** | 85.56 |

## 5.1 Datasets and Baselines

**Datasets.** We follow the inductive link prediction in the basic model GraIL [31] which proposes benchmarks derived from WN18RR [9], FB15K-237 [32] and NELL-995 [40], and each has been divided into four versions. Each version of a dataset consists of a pair of KGs named *train-graph* and *ind-test-graph*, whose entities are without intersection. Each *train-graph* has its valid KG for evaluation during training. The statistics of benchmark datasets is illustrated in Table 6 in the Appendix.

**Baselines.** For comparison, we select typical models for the inductive relation prediction task, including rule-based RuleN [21], Neural-LP [43] and DRUM [27], and graph-based GraIL [31], CoM-PILE [19], TACT [5], and recent works RED-GNN [49], LogCo [25], and RMPI [14]. For a fair comparison, all the results of baselines are from published papers, which are implemented by a consistent link prediction setting.

## 5.2 Metrics and Experimental Settings

**Metrics.** In the comparison and other tasks for evaluating the performance of SymRITa, we implement both classification and ranking metrics for multiple runs considering the random seeds and samples. For the classification task, AUC-PR is an indicator to calculate the area under the prediction-recall curve and it evaluates if the triple is valid. In order to calculate the AUC-PR, we generate a negative triple by replacing the head or tail with a random entity of each positive triple in the test set. For the ranking metric Hits@10, we evaluate it in a general link prediction mode by ranking the score of test triples among 50 random negative samples, and see if the true triple can rank in the top 10.

**Experimental Details.** For the subgraph extraction, we obtain 3-hop enclosing subgraphs by the double vertex labeling. In the graph embedding process, we employ a 3-layer GCN with the dimension as 64. For the logic transformer, we set the dimension of the hidden state as 256, the number of heads as 6, and the number of layers as 2. For the hyper-parameters $\lambda$, we set it to 0.8 and illustrate the reason in Section 5.5. During the training process, we set batch size as 16 and we use Adam [17] as the optimizer with learning rate being 0.0005. The maximum length of rules are set as $L_{max} = 2, 3$ and the margin value in the training loss is 5. We implement the experiments on one NVIDIA's Tesla V100 graphic card. More detailed settings and the reproducibility are in Appendix B.2 and B.3.

## 5.3 Comparison Results

In this subsection, we use the two metrics on the classification and ranking tasks respectively to illustrate the effectiveness of SymRITa. SymRITa is evaluated on twelve inductive benchmarks of KGs from [31]. The reasoning results are shown in Table 2. All the results of baselines are from published papers on top conferences.

For the **classification task** reflected by the metric AUC-PR, SymRITa can obviously outperform all the selected state-of-the-art baselines, since it obtains optimal or suboptimal values of AUC-PR on all the datasets in Table 2. Specifically, except for WN18RR_v4, SymRITa outperforms all the listed inductive relation prediction methods on the classification task. Even on WN18RR_v4, SymRITa gets the suboptimal value of AUC-PR, which is 0.74% slightly lower than the SOTA method CoMPILE. Nevertheless, the average boost of SymRITa on WN18RR compared to CoMPILE is 0.28%. As for the recent work RMPI-NE-TA, SymRITa outperforms it on all the datasets with the AUC-PR values. All the phenomena demonstrate

**Table 3: Ablation results on inductive benchmarks derived from FB15K-237_v1 and NELL-995_v1.**

| Method | FB15K-237_v1 | | | | | NELL-995_v1 | | | | |
|---|---|---|---|---|---|---|---|---|---|---|
| | MRR | H@1 | H@5 | H@10 | AUC-PR | MRR | H@1 | H@5 | H@10 | AUC-PR |
| SymRITa | **51.98** | **42.43** | **62.20** | **72.20** | **89.77** | **50.34** | **44.00** | **56.00** | **62.50** | **92.59** |
| SymRITa w/o Symbolic Rules | 48.68 | 39.75 | 57.07 | 65.61 | 86.01 | 47.56 | 42.70 | 48.80 | 54.20 | 83.53 |
| Δ | ↓3.30 | ↓2.68 | ↓5.13 | ↓6.59 | ↓3.76 | ↓2.78 | ↓1.30 | ↓7.20 | ↓8.30 | ↓9.06 |
| SymRITa w/o Logic Transformer | 48.56 | 38.54 | 58.04 | 66.34 | 86.01 | 48.11 | 42.50 | 53.50 | 58.50 | 85.27 |
| Δ | ↓3.42 | ↓3.89 | ↓4.16 | ↓5.86 | ↓1.90 | ↓2.23 | ↓1.50 | ↓2.50 | ↓4.00 | ↓7.32 |
| SymRITa w/o Conjunction Attention | 49.77 | 40.00 | 60.98 | 69.26 | 88.04 | 49.92 | 43.50 | 53.00 | 60.10 | 87.85 |
| Δ | ↓2.21 | ↓2.43 | ↓1.22 | ↓2.94 | ↓1.73 | ↓0.42 | ↓0.50 | ↓3.00 | ↓2.40 | ↓4.74 |

the effectiveness of SymRITa in predicting whether the triple is valid with unseen entities.

As for the **ranking task** reflected by Hits@10, SymRITa can also achieve competitive effectiveness compared to the baselines on twelve datasets. Nine out of twelve Hits@10 values are better than other baselines. Other methods with inductive ability, whether rule-based or graph-based, can not obtain the all-round superiority. SymRITa obtains as much as 2.98%, 1.69%, 7.60% average performance improvements in Hits@10 on WN18RR, FB15K-237 and NELL-995 respectively compared to RMPI-NE-TA, which is the most recent inductive relation prediction work. For the ranking results which are slightly lower, the optimal results are distributed in different categories of methods, instead of being achieved by a prominent one.

Overall, the results indicate the effectiveness and development of SymRITa, which integrates the symbolic rules in acquiring inductive ability in KGs.

## 5.4 Ablation Studies

We investigate the impacts of rules and the logic transformer in obtaining the inductive ability of the model. As shown in Table 3, we rerun the model without the factors on FB15K-237_v1 and NELL-995_v1 respectively, and the methods are denoted as:

- **SymRITa w/o Symbolic Rules** indicates the method removing the constructed rules in the subgraph.
- **SymRITa w/o Logic Transformer** indicates the method removing logic transformer in the subgraph. To implement this model, we replace the logic transformer with the mean operator. It can be regarded as the effectiveness of the logic transformer and the conjunction attention simultaneously.
- **SymRITa w/o Conjunction Attention** indicates the method removing the conjunction attention in the logic transformer.

The ablation results in Table 3 illustrate the effectiveness of the significant factors in SymRITa. **(1) The inductive performance achieves optimal results when all components work simultaneously.** In both FB15K-237_v1 and NELL-995_v1, the integrating of three factors helps SymRITa obtain obvious classification and ranking improvements. **(2) The extracted rules from the subgraph are essential in inductive relation prediction.** From the AUC-PR and link prediction results of SymRITa w/o Symbolic Rules, the reductions indicate the significance of incorporated rules in the model. Specifically, compared to the original SymRITa, SymRITa w/o Symbolic Rules reduces the prediction results by 4.29% and

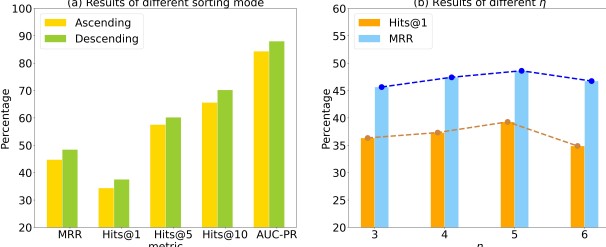

**Figure 4: Effectiveness of the sorting mode and the margin $\eta$.**

5.73% on two datasets respectively. The main reason is that the implicit logics contained in the rules help improve the inductive ability. **(3) The logic transformer and conjunction attention in it are also critical in SymRITa.** The prediction performance of SymRITa w/o Logic Transformer averagely drops by 3.85% and 3.51% on FB15K-237_v1 and NELL-995_v1, respectively. The reduction of SymRITa w/o Conjunction Attention is slighter, but the average decline compared to SymRITa still illustrates its effectiveness.

## 5.5 Weight Analysis

**Analysis of the sorting mode.** During training, the sorting mode of the sorter is critical in the rule integration process. It decides the order of rules in the same subgraph. We record the classification and ranking results with an ascending sorter and a descending sorter respectively on FB15K-237_v1. The results are shown in Figure 4(a). It can be figured that the descending order benefits both classification and ranking tasks. The phenomenon is consistent with the analysis in Section 4, that the more important rule with its corresponding position information can precisely embed the symbols of rules in a subgraph.

**Analysis of the margin $\eta$.** In SymRITa, we use a parameter $\eta$ in the loss function to optimize the training process. In the loss function, it decides the gap between the positive and negative samples. The results on FB15K-237_v1 are in Figure 4(b). From the results of classification metric AUC-PR and ranking metric Hits@1, when $\eta = 5$, SymRITa obtains the most effective performance. It indicates that a large or small margin will lead to a negative impact.

**Analysis of hyper-parameter $\lambda$.** In SymRITa, $\lambda$ controls the conjunction attention weight in the logic transformer when modeling the rules. We rerun the training process on FB15K-237_v1 and NELL-995_v1, and record the results of classification and ranking tasks in different values of $\lambda \in [0.1, 1.2]$ with a step of 0.1. The results of AUC-PR and MRR are shown in Figure 5. From the distributions of results on FB15K-237_v1, we observe that the inductive

**Table 4: The illustration of the effectiveness of SymRITa by several real cases. The output rules and their corresponding confidences are generated by SymRITa and the model without logic transformer named "w/o LT".**

| Subgraph | head← | body | $\beta$ SymRITa | w/o LT |
|---|---|---|---|---|
| 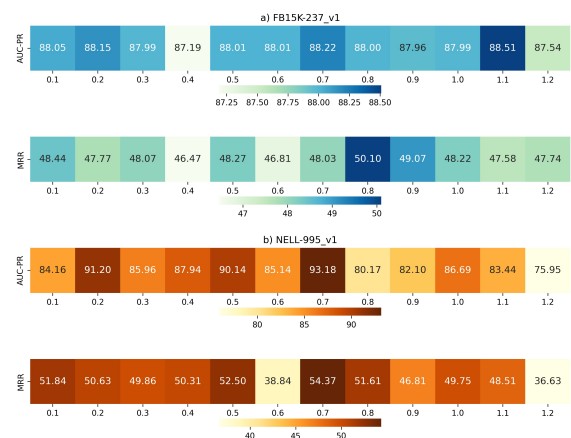 | $subpart\_of\_organization(X, Y) \leftarrow$ | $top\_member\_of\_organization(X, Y)$ | **1** | 0.9708 |
| | $subpart\_of\_organization(X, Y) \leftarrow$ | $company\_economic\_sector(X, Z) \wedge subpart\_of(Z, Y)$ | **<0.0001** | 0.0292 |
| | $agent\_controls(X, Y) \leftarrow$ | $company\_economic\_sector(X, Z) \wedge agent\_controls(Z, Y)$ | **1** | 0.1579 |
| | $agent\_controls(X, Y) \leftarrow$ | $subpart\_of\_organization(X, Z) \wedge agent\_controls(Z, Y)$ | **<0.0001** | 0.8421 |
| | $agent\_controls(X, Y) \leftarrow$ | $organization\_headquartered\_in\_city(X, Y)$ | **<0.0001** | <0.0001 |

a) FB15K-237_v1

AUC-PR | 88.05 | 88.15 | 87.99 | 87.19 | 88.01 | 88.01 | 88.22 | 88.00 | 87.96 | 87.99 | 88.51 | 87.54
0.1 0.2 0.3 0.4 0.5 0.6 0.7 0.8 0.9 1.0 1.1 1.2
87.25 87.50 87.75 88.00 88.25 88.50

MRR | 48.44 | 47.77 | 48.07 | 46.47 | 48.27 | 46.81 | 48.03 | 50.10 | 49.07 | 48.22 | 47.58 | 47.74
0.1 0.2 0.3 0.4 0.5 0.6 0.7 0.8 0.9 1.0 1.1 1.2
47 48 49 50

b) NELL-995_v1

AUC-PR | 84.16 | 91.20 | 85.96 | 87.94 | 90.14 | 85.14 | 93.18 | 80.17 | 82.10 | 86.69 | 83.44 | 75.95
0.1 0.2 0.3 0.4 0.5 0.6 0.7 0.8 0.9 1.0 1.1 1.2
80 85 90

MRR | 51.84 | 50.63 | 49.86 | 50.31 | 52.50 | 38.84 | 54.37 | 51.61 | 46.81 | 49.75 | 48.51 | 36.63
0.1 0.2 0.3 0.4 0.5 0.6 0.7 0.8 0.9 1.0 1.1 1.2
40 45 50

**Figure 5: Effectiveness evaluation by MRR (%) and AUC-PR(%) of parameters $\lambda$ over two datasets.**

performance varies with the different values of $\lambda$, indicating the weight of conjunction attention in the logic transformer. For the classification task, SymRITa obtains better AUC-PR results when $\lambda > 0.7$ or $\lambda \leqslant 0.2$, while it achieves better MRR result when $\lambda = 0.8$ in the ranking task. On NELL-995_v1, SymRITa obtains optimal results on both classification and ranking tasks when $\lambda = 0.7$. Therefore, we select $\lambda = [0.7, 0.8]$ when implementing the experiments.

## 5.6 Case Studies

As we illustrate in Section 3, the rule from KGs is in a form of the first-order logic with its corresponding confidence $\beta$. In Table 4, it illustrates the first-order logic rules with the confidences of SymRITa and the model without the essential factor logic transformer on NELL-995_v1. SymRITa can adjust the importance of each logic rule by obtaining precise embeddings of rules in a subgraph. For instance, if the target relation is *agent_controls*, the two rules in the same subgraph are shown in the last row of Table 4. The logic transformer increases the significance of rule *agent_controls* ← *company_economic_sector*$(X, Z) \wedge$ *agent_controls*$(Z, Y)$, which is reflected by the improvement of $\beta$ from 0.1579 to 1. In reality, we can figure that the rule *agent_controls*$(X, Y) \leftarrow$ *subpart_of_organization*$(X, Z) \wedge$ *agent_controls*$(Z, Y)$ will not be reasonable in reasoning the query $(h, agent\_controls, t)$? where *agent_controls* is the target relation. In most situation, the subpart could not represent the entire organization, which would lead to a incorrect reasoning result.

**Table 5: Ranking results of MRR (%) on inductive benchmarks with unseen relations derived from NELL-995.v1.v3, NELL-995.v4.v3 and FB15k-237.v1.v4**

| Method | NELL-995.v1.v3 | NELL-995.v4.v3 | FB15k-237.v1.v4 |
|---|---|---|---|
| TACT-base | 43.59 | 52.68 | 61.02 |
| RMPI-base | 59.10 | 70.33 | 56.81 |
| RMPI-NE | 56.19 | 59.47 | 57.77 |
| SymRITa | **65.54** | **70.71** | **61.64** |

## 5.7 Unseen Relation Prediction

In addition, we evaluate the generalized ability on the inductive settings with not only unseen entities but also unseen relations. This is a more challenging setting proposed in [14], which is designed for the unseen relations during testing. The detailed statistics of the datasets can be found in the Appendix B.1.

We record the ranking results of MRR on three test datasets in Table 5. The ranking results outperform the baselines on three inductive datasets. The performance indicates that the symbolic rule integration helps SymRITa to deal with the issue with several unseen relations mixed in the test set.

## 6 CONCLUSION AND FUTURE WORK

In order to solve the entity independence of inductive relation prediction in KGs, we propose a method named SymRITa to integrate symbolic rules during the reasoning process. First, we extract subgraphs based on the target triple, and rules from each subgraph. Second, we obtain the subgraph-based embeddings by a GCN and symbol-based embeddings by the logic transformer with a conjunction attention mechanism. The logic transformer is designed for the challenges of argument variability and predicate non-commutativity in modeling rules and preserving their precise representations. Finally, SymRITa captures the inductive ability by combining subgraph-based and symbol-based embeddings, and jointly trains the model. The experiments on twelve inductive datasets show the effectiveness of SymRITa, and comprehensively demonstrate the impacts of rules and the significance of factors from our model.

SymRITa still needs improving in terms of performance and scalability. We would like to extend the symbolic rules in which the atoms are connected by disjunction, and expand the inductive scenario to commonsense knowledge graphs [30].

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

**Table 6: Statistics of Inductive Relation Prediction Datasets.**

|  | WN18RR | | | FB15K-237 | | | NELL-995 | | |
|---|---|---|---|---|---|---|---|---|---|
|  | #R | #E | #Tr | #R | #E | #Tr | #R | #E | #Tr |
| v1 | 9 | 2,746 | 6,678 | 183 | 2,000 | 5,226 | 14 | 10,915 | 5,540 |
| v1-ind | 9 | 922 | 1,991 | 146 | 1,500 | 2,404 | 14 | 225 | 1,034 |
| v2 | 10 | 6,954 | 18,968 | 203 | 3,000 | 12,085 | 88 | 2,564 | 10,109 |
| v2-ind | 10 | 2,923 | 4,863 | 176 | 2,000 | 5,092 | 79 | 4,937 | 5,521 |
| v3 | 11 | 12,078 | 32,150 | 218 | 4,000 | 22,394 | 142 | 4,647 | 20,117 |
| v3-ind | 11 | 5,084 | 7,470 | 187 | 3,000 | 9,137 | 122 | 4,921 | 9,668 |
| v4 | 9 | 3,861 | 9,842 | 222 | 5,000 | 33,916 | 77 | 2,092 | 9,289 |
| v4-ind | 9 | 7,208 | 15,157 | 204 | 3,500 | 14,554 | 61 | 3,294 | 9,520 |

## A  TRAINING PROCEDURE

SymRITa can predict missing relations of incomplete triples in an inductive setting, and also integrate first-order rules for explanation at the same time. In Algorithm 1, we demonstrate the process of predicting relations by SymRITa. SymRITa uses a KG as the input and at last outputs the score of target triple and a set of first-order rules with confidences.

---

**Algorithm 1** Process of integrating rules by SymRITa

---

**Input:** KG $G \langle R, E, T \rangle$, target triple $a_T$, hyper-parameters $\lambda, \eta$, etc.
**Output:** Score of $a_T$ and First-order logic rules.
1: Extract subgraph $\mathcal{S}_T$ around each $a_T$, and initialize each node.
2: **for** each training iteration **do**
3:   **for** each batch of triples in $G$ **do**
4:     Obtain embeddings of entities and relations in $\mathcal{S}_T$.
5:     Extract rules within the length $L_{max}$ and generate symbolic rule graphs.
6:     $\mathcal{F} \leftarrow$ Sort the rules with values of $\theta$.
7:     $F \leftarrow$ Get input of the logic transformer.
8:     $F_o \leftarrow$ Add position information PosEmb($F$) in the symbolic rule graphs.
9:     $M_{conj} \leftarrow$ Obtain the matrix in the conjunction attention by symbolic rule graphs.
10:     Calculate self-attention with matrices $Q, K, V$ and $M_{conj}$ by Eq. (10) and Eq. (9).
11:     $H^{(L_2)} \leftarrow$ Output the logic transformer.
12:     Get subgraph-based embedding $S$ and symbol embedding $P$ with confidence of each rule $\beta$.
13:     Score $a_T$ combining $S$ and $P$.
14:     $\mathcal{L} \leftarrow$ Get margin-based loss by Eq. (17).
15:     Update the parameters by Adam optimizer.
16:   **end for**
17: **end for**
18: **return** Score of $a_T$ and first-order rules in $\mathcal{F}$ with $\beta$.

---

## B  EXPERIMENTAL DETAILS

### B.1  Datasets

The statistics of benchmark datasets is illustrated in Table 6 in the Appendix. These datasets consist of train and test KGs with the inductive setting. Each version of a dataset consists of a pair of KGs named *train-graph* and *ind-test-graph*, whose entities are without intersection. Meanwhile, *train-graph* contains all the relations in *ind-test-graph*. Each *train-graph* has its valid KG for evaluation during training.

**Table 7: Statistics of Datasets with Unseen Relations in the Test Set. The numbers in the brackets are the numbers of unseen relations.**

|  | NELL-995.v1.v3 | | | NELL-995.v4.v3 | | | FB15k-237.v1.v4 | | |
|---|---|---|---|---|---|---|---|---|---|
|  | #R | #E | #Tr | #R | #E | #Tr | #R | #E | #Tr |
| train | 14 | 3103 | 5540 | 76 | 2092 | 9289 | 180 | 1594 | 5226 |
| test | 106 (98) | 2271 | 5550 | 110 (53) | 3140 | 8308 | 200 (26) | 3051 | 14554 |

For the datasets with unseen relations in [14], the detailed statistics is in Table 7. These datasets are denoted using the pattern "XXX.v$i$.v$j$", where XXX is the source transductive dataset. $i$ is the index indicating which version of inductive benchmark the training graph comes from, while $j$ is the version of the testing graph.

### B.2  Experimental Settings

Different parameters might influence the performance on different datasets, so the parameters are tuned separately. We conduct experiments in the following search space of parameters:

- Learning rate: {0.0001, 0.0002, 0.0005, 0.001}
- The number of subgraph hops: {2, 3, 4}
- Maximum length of relational paths $L_{max}$: {1, 2, 3}
- Margin hyper-parameter $\eta$: {3, 4, 5, 6}
- Number of heads in logic transformer: {5, 6, 7, 8}
- Number of layers in logic transformer: {2, 3, 4}

### B.3  Code Appendix

For reproducibility, core codes of SymRITa are in an anonymous hyperlink: SymRITa. In the future, we will make all the available source codes open for the method upon publication of the paper.

Received 20 February 2007; revised 12 March 2009; accepted 5 June 2009

