# OpenReview forum: "A Symbolic Rule Integration Framework with Logic Transformer for Inductive Relation Prediction"
_ACM.org/TheWebConf/2024/Conference — TheWebConf24_

### Official Review · Reviewer_FyCj · 2023-11-10

**Novelty:** 5
**Technical Quality:** 6

**Review:**

The authors of this article present SymRITa, an architecture that integrates symbolic (Horn) rules into a logic transformer for the sake of inductive relation prediction in knowledge graphs. This is the task of predicting the relation between two unseen entities. SymRITa first learns rules from sub-graphs extracted around the target entities, which are then embedded alongside the KG entities using a logic transformer that implements a conjunction attention mechanism. An experimental evaluation compares the performance of SymRITa to SOTA methods on triple classification and relation link prediction using standard benchmarks. The results are rather encouraging and suggest that the proposed approach generally outperforms existing methods, including symbolic rule-based, neurosymbolic rule-based, and graph-based approaches. The evaluation includes an ablation study that assess the contribution of each of SymRITa's components to the overall performance.

Overall this is a nice paper with a clear and interesting contribution and positive results. While I lean towards acceptance, I would appreciate if the authors clarified a few things in regards to the method and the experimental evaluation.

- In Section 4.2.2 is not clear to me how variable order is handled in the conjunction attention matrix. What if the conjunctive expression in Fig. 3 were r_l(Z1, Z2) ^ r5(Z1, X) ^ r7(Z2, Y)? Is there a canonical form for the rules?

- At the end of the paper the authors suggest that runtime is still significant. Would it be possible to hint why is it the case? I am asking this because as far as I understand, SymRITa extracts rules from subgraphs constructed around each of the training triples. There may be a significant overlap between those sub-graphs. Would it be possible to optimize the rule extraction by "mutualizing" it among different triples? In that case the authors could benefit from existing, more optimized, rule mining methods such as AnyBurl or AMIE.

- The paper's research perspectives in the conclusion make sense to me, however I think there is an elephant in the room. How well does SymRITa perform in the transductive scenario? A small subsection about this would have been interesting, albeit not compulsory.

- Section 5.7 hints results about inductive reasoning for unseen relations, but honestly, I do not understand how exactly this could be achieved with the current architecture. I think the authors are obliged to explain the experimental setup a bit more and provide a few results -- pointing to the Appendix does not make sense, specially if the task is not properly explained. If space is an issue (which I understand), the authors could just provide a few key numbers in the text and refer to the supplementary material for more details.

- How does exactly the model work? I give an input as query, e.g., a triple to classify, two entities with missing relation, and then the system provides a ranking with rules (to which we have attached a confidence score)? A couple of lines about this would be appreciated.

**Questions:**

- When training the model, how are negative examples computed?
- In Section 4.2.2 is not clear to me how variable order is handled in the conjunction attention matrix

**Ethics Review Description:**

No issue

**Reviewer Confidence:**

2: The reviewer is willing to defend the evaluation, but it is likely that the reviewer did not understand parts of the paper

**Scope:**

3: The work is somewhat relevant to the Web and to the track, and is of narrow interest to a sub-community

---

### Official Review · Reviewer_Dcit · 2023-11-24

**Novelty:** 6
**Technical Quality:** 6

**Review:**

This paper addresses the incorporation of path-form rules to enhance knowledge graph completion. The authors propose using conjunctive attention to model the structural information of rules and introduce a logic transformer to unify the modeling of both knowledge graph structure and terms.

Pros
1. The proposed way of modeling symbolic rules is novel.
2. This paper is well formatted and easy to read.

Cons
1. The selection of evaluation metrics may need further discussion.
2. The effectiveness of logic extraction and sorter require additional analysis.

**Questions:**

1. Table 2 only reports Hit@K for K=10. How about the cases for K=1 or K=5? Comparisons with other values of K should be provided in the appendix at a minimum.

2. Can the authors provide some additional analyses about the sorting results of the constructed implicit rules? The paper suggests that modeling rules should reflect predicate non-commutativity, but the similarity function $\theta$ seems not to consider the order of predicates. Is this an inconsistency?

3. Is the performance of the proposed method sensitive to the order in which implicit rules are extracted? Could the authors provide an in-depth explanation of this?

**Reviewer Confidence:**

3: The reviewer is confident but not certain that the evaluation is correct

**Scope:**

3: The work is somewhat relevant to the Web and to the track, and is of narrow interest to a sub-community

---

### Official Review · Reviewer_ytBV · 2023-11-30

**Novelty:** 6
**Technical Quality:** 6

**Review:**

Recap:
The authors propose a new machine-learning architecture for link prediction between entities not in the training set. As a side product, it produces confidence scores for Horn rules, which are meant to explain the system's reasoning.

Quality:
The system beats all the baseline systems cited on most of the datasets shown, albeit sometimes by a very small margin.

Clarity:
The paper is moderately hard to read, due to unexplained technical words, especially in the introduction and related work.

Originality:
According to the authors, no one has used both topological link prediction and rule-based link prediction in one system before. However, I am not competent to judge how much novelty there is in this, since I am not familiar with the related work.

Significance:
This approach gets some significance just for achieving state-of-the-art performance, but I think that its other novel feature, explainability via rules, is not particularly significant, since there is no way of knowing what the significance scores assigned to the rules mean. I don't see any guarantee that the system is actually using the rules it has mined as rules, to make its predictions.

Pros:

State-of-the-art performance

Interesting case studies showing correct classification of a few rules as high-confidence correct or high-confidence incorrect

Cons:

Paper does not explain the architecture in full, one would have to look at the code (e.g., what is the positional embedding used? How is the embedding of r obtained? What part of the logic transformer contains the 'hidden states' used as its output?)

Explainability feature is not really explanatory

Comments: The rules examined by the system are all of the form $\bigwedge\limits_1^{n-1} A_i(X_i,X_{i+1}) \to B(X_1,X_n)$. Although it's not clear how the semantics of Horn rules is related to what the system does anyway, it might be worth pointing out that this is a less general form than Horn rules - i.e. not every Horn rule is equivalent to a set of rules in this form. For example, $A(X,Y) \land A(Y,Z) \land A(Z,X) \to B(X,Y)$.

Underlines are missing in some columns of Table 2.

'Prediction-recall curve' - should be 'precision-recall curve'?

------------

Having read the rebuttal, I've increased my scores on novelty and relevance to the track.

**Questions:**

1. What is unscathed conjunction attention?
2. I'm confused about 'target triple'. In 3.1 it seems to refer to a known triple in the training graph, but elsewhere in the paper 'target triple' refers to a triple whose presence in the test graph is to be predicted.
3. In Equation 3, omega_i,r does not have a j index. Which v_j is being referred to on the right-hand side?
4. Equation 17: What is the positive sample a_T^+ for a given triple a_T?
5. What is the positional embedding you used?
6. Where does the embedding of r come from, as used e.g. in Equation 3?
7. What part of the logic transformer contains the 'hidden states' H(L_2)?

**Ethics Review Description:**

No issue

**Reviewer Confidence:**

2: The reviewer is willing to defend the evaluation, but it is likely that the reviewer did not understand parts of the paper

**Scope:**

3: The work is somewhat relevant to the Web and to the track, and is of narrow interest to a sub-community

---

### Official Review · Reviewer_Th2H · 2023-12-01

**Novelty:** 7
**Technical Quality:** 5

**Review:**

## Summary
The paper proposes a method for inductive link prediction on Knowledge Graphs. It uses a model consisting of a Graph Neural Network to learn node embeddings, and a Rule Transformer to embed rules extracted from a subgraph. This subgraph is a n-hop graph around the target triple to be predicted. The node embeddings of the subgraph as well as the rule embeddings are aggregated and combined into a final representation, which is used to predict the link of interest.

## Strong Points
S1. The paper tackles a relevant problem in knowledge graphs.
S2. The paper provides an interesting step into interpretable link prediction, as explicit first-order rules are mined and a confidence parameter beta, which indicates the importance of a given rule for the link prediction, is returned.
S3. The paper shows promising experimental results on established benchmarks compared to state-of-the-art methods.

## Weak Points
W1. The paper presents the architecture in great detail. However, some specifics are still unclear, and some appear not thoroughly ablated (like ordering the rules and incorporation of r_T into every GNN embedding).
W2. The scalability of the model is not studied or discussed.

## Detailed Comments
- The GNN equations in equations 2, 3, and 4 are referred to as convolutional, however they represent a Graph Attention Network.

- It is unintuitive for me that all mined rules from a subgraph are simultaneously given to the Rule Transformer. What is the rational of letting nodes from different rules interact inside of a self-attention system? If the rules are supposed to provide a signal for an existing link by themselves, they should be processed in isolation. This also holds true for the subsequent fusion. This makes sure that the model really relies on a single rule for link prediction, and not the complete context of the topology of the subgraph. I understand that this is in part compensated by Eq.15, but that is only ensuring a soft focus, and as stated, the representation of the individual rules itself are superpositions of all the mined rules for the subgraph.

- In regards to the above, i.e. that the complete topology of nodes and edges are embedded (although via two different networks), what is the expressive difference to a single edge-enhanced GNN (like R-GCN etc.)? Since the outputs of the node GNN and the rule transformers are pooled (Eq. 12,13, 14,15) and aggregated (Eq.16), this could equally represented by a edge-enhanced GNN. Although this does not allow to assign a weight to mined first-order rules it could be more expressive since a more detailed representation of the topology with regard to connectivity of nodes and edges would be provided. In that regard, I think a comparison to such models in the experimental section is necessary.

- As far as in understand, given a query h, r_T, t, the representation v_T in Eq.12 is the same for every entity in the subgraph. That is, the model would predict the same link probability for all nodes in the subgraph. Why is here not just the embedding for the head h used ? In general, I dont see the head h and tail t of interest appearing in the equations, which leads me to the believe that the same confidence for a given relation r_T would be assigned to each combination of two nodes from the subgraph. I understand that in general, if another combination is chosen, a different subgraph would be found, but I assume that there can be situations where the subgraphs would be the same.

- Furthermore, Line 543 states tht v_aT is the concatenated embedding of entities. Is by concatenation aggregation meant ? Because this is also happening in v_T (Eq.12). If not, it is unclear to my why concatenation and mean aggregation are used both. Additionally, If a concatenation of embeddings of all nodes in the subgraph is used, the vector S (Eq.13) would be of variable length, hence It would not be possible to multiply with a fixed size matrix W_score in Eq. 16.

- The result of 5.5, analysis of the sorting mode is counterintuitive to me. Since an encoding based Transformer Architecture is permutation equivariant with respect to its input, a complete reversal of the order of the inputed rules should make no difference (which is displayed in figure 4a).

- The paper does not discuss potential limitations of the approach in terms of scalability. To get a prediction for a potential link h r_T t, the full embedding of the GNN, as well as the Rule Transformer (because r_T influences the ordering of the rules (5)) need to be calculated for every possible assignment of r_T. This is not scalable for larger graphs. Existing methods often try to separate embeddings so that link prediction can be performed by a simple vector multiplication instead of running a complete model for each possible combination. As already stated in the Conclusion, there needs to be an architectural change to make this more scalable (like decoupling r_T from the rule transformer and GNN).

**Questions:**

Q1. The method appears to only mine path-based conjunctive first-order rules. Why are no other forms incorporated in the experiments?  (e.g. star shaped etc.)

Q2. How are edge embeddings learned? Are they learned by backpropagating through the Rule Transformer ? Or are they learned by means of the GNN (Eq. 3, 4)?

Q3. In Line 733 of the Ablation Study, if symbolic rules are removed but not the Logic Transformer, what is the input to the logic transformer?

**Ethics Review Description:**

--

**Reviewer Confidence:**

2: The reviewer is willing to defend the evaluation, but it is likely that the reviewer did not understand parts of the paper

**Scope:**

3: The work is somewhat relevant to the Web and to the track, and is of narrow interest to a sub-community

---

### Decision · Program_Chairs · 2024-01-22

**Decision:**

Accept

**Comment:**

/ Scope is limited (consistently "narrow sub-field")
 High novelty and technical quality -- and corresponding confidence -- with only requests for a few clarifications that authors should strongly consider to incorporate directly into their narrative, or at least appendix.